# Building accurate sequence-to-affinity models from high-throughput in vitro protein-DNA binding data using FeatureREDUCE

**Todd R Riley[1,2,3], Allan Lazarovici[1,4], Richard S Mann[2,5], Harmen J Bussemaker[1,2]\***

[1]Department of Biological Sciences, Columbia University, New York, United States; [2]Department of Systems Biology, Columbia University, New York, United States; [3]Department of Biology, University of Massachusetts Boston, Boston, United States; [4]Department of Electrical Engineering, Columbia University, New York, United States; [5]Department of Biochemistry and Molecular Biophysics, Columbia University, New York, United States

**Abstract** Transcription factors are crucial regulators of gene expression. Accurate quantitative definition of their intrinsic DNA binding preferences is critical to understanding their biological function. High-throughput in vitro technology has recently been used to deeply probe the DNA binding specificity of hundreds of eukaryotic transcription factors, yet algorithms for analyzing such data have not yet fully matured. Here, we present a general framework (FeatureREDUCE) for building sequence-to-affinity models based on a biophysically interpretable and extensible model of protein-DNA interaction that can account for dependencies between nucleotides within the binding interface or multiple modes of binding. When training on protein binding microarray (PBM) data, we use robust regression and modeling of technology-specific biases to infer specificity models of unprecedented accuracy and precision. We provide quantitative validation of our results by comparing to gold-standard data when available.

**\*For correspondence:** hjb2004@columbia.edu

**Competing interests:** The authors declare that no competing interests exist.

## Introduction

Transcription factors (TFs) play a central role in the regulation of gene expression. To be able to understand and predict the behavior of the gene regulatory circuitry in any given organism, we need to know the in vivo DNA binding preferences of the TFs that its genome encodes. In recent years, a number of high-throughput in vitro technologies have been introduced that can provide such information (*Berger and Bulyk, 2009*; *Warren et al., 2006*; *Maerkl and Quake, 2007*; *Zhao et al., 2009*; *Slattery et al., 2011*; *Berger et al., 2006*). However, while the volume of the data generated using these assays dwarfs that of more traditional measurements of protein-DNA interaction strength, the available computational methodology for analyzing them has not fully matured (*Weirauch, 2013*).

The number of base pairs that constitute the DNA 'footprint' within which base identity can influence binding affinity depends strongly on the three-dimensional structure of the DNA-binding domain (DBD) of the TF. In theory, as long as thermodynamic equilibrium can be assumed, sequence specificity is completely defined by a table containing the (relative) affinity with which the DBD binds to each possible oligonucleotide within the footprint. This tabular approach has been widely used to analyze Protein Binding Microarray (PBM) data (*Berger and Bulyk, 2009*). It comes with significant challenges, however. First, the size of the oligomer table grows exponentially with footprint size,

**eLife digest** Transcription is the process by which the information contained within DNA is copied to a short-lived molecule called RNA so that it can be transported to other areas of the cell for various purposes. Transcription factors are key components in this process. These proteins recognise and gather at specific sequences of DNA near genes, and then assist the enzymes that copy the information in the gene into a molecule of RNA. This means that transcription factors essentially control which genes are expressed, and when and where these genes are expressed.

Recent technological advances have made it possible to identify where transcription factors can bind within DNA sequences. Yet, while a lot of data has been generated in this area, the computational tools needed to make sense of it have not kept pace.

Now, Riley et al. have developed software called FeatureREDUCE that will allow researchers to build computer predictions of how strongly a transcription factor will interact with specific short sections of DNA sequence. The software can be applied to experimental data collected in so-called 'protein binding microarray' experiments. FeatureREDUCE can also be used to investigate questions in the field of transcription factor research that had previously remained unanswered. First, to what level of detail can data obtained from recent technological advances be understood? Second, can transcription factors bind to DNA in more than one way, and can data from protein binding microarrays be used to uncover this?

Riley et al. show that FeatureREDUCE can produce accurate and interpretable clues about the biology behind how transcription factors recognize DNA sequences. This includes how mutations as small as a change to single DNA letter can affect recognition. The next step will be to use the software to make sense of the existing volume of experimental data regarding protein-DNA interactions and data that will be generated in future experiments.

which in practice limits it to eight base pairs, shorter than the footprint of most TFs. Even for octamer tables, the number of affinity parameters to be estimated is on the same order as the number of PBM data points. This limits precision and necessitates the use of non-parametric methods (as opposed to parameterized biophysical methods), resulting in an associated loss of quantitative information.

A long-standing alternative has been to assume that each nucleotide position within the footprint contributes independently to the overall binding affinity. The most commonly used representation of sequence specificity that makes this independence assumption is the position weight matrix (PWM) (*Berg and von Hippel, 1987*; *Stormo and Fields, 1998*; *Stormo, 2000*; *Djordjevic et al., 2003*), which defines position-specific base frequencies. Algorithms for inferring the PWM coefficients traditionally aim to maximize its information content relative to a random background model (*Lawrence et al., 1993*; *Frech et al., 1997*; *Bailey, 1995*; *Roth et al., 1998*). In an alternative approach, sequence specificity is represented in terms of the relative affinity (or, equivalently, the difference, $\triangle\triangle G$, in binding free energy) associated with each possible point mutation of the optimal sequence (*Stormo and Yoshioka, 1991*; *Stormo et al., 1993*), and summarized in the form of a position-specific affinity matrix (PSAM) (*Foat et al., 2006*; *Bussemaker et al., 2007*). The difference in philosophy between the PWM and PSAM representations also leads to a different approach to estimating their coefficients. It is no longer the information content (i.e. the height of the letters in the standard sequence logo) that is being optimized, but rather the ability of the PSAM to quantitatively explain the variation in a measurable quantity in terms of variation in the nucleotide sequence associated with each quantity (for instance, the expression level of a gene in terms of its upstream promoter sequence). In the case of PBM data, the PSAM parameters are inferred by performing a nonlinear fit of a sequence-based model that predicts the signal intensity for each probe. The first implementations of this idea were the *MatrixREDUCE* (*Foat et al., 2006*; *Foat et al., 2005*) and *PREGO* (*Tanay, 2006*) algorithms; a more recent extension is *BEEML-PBM* (*Zhao and Stormo, 2011*).

Whether dependencies between nucleotide positions can be accurately estimated from PBM data and used to refine models of binding specificity remains an open question (*Weirauch, 2013*;

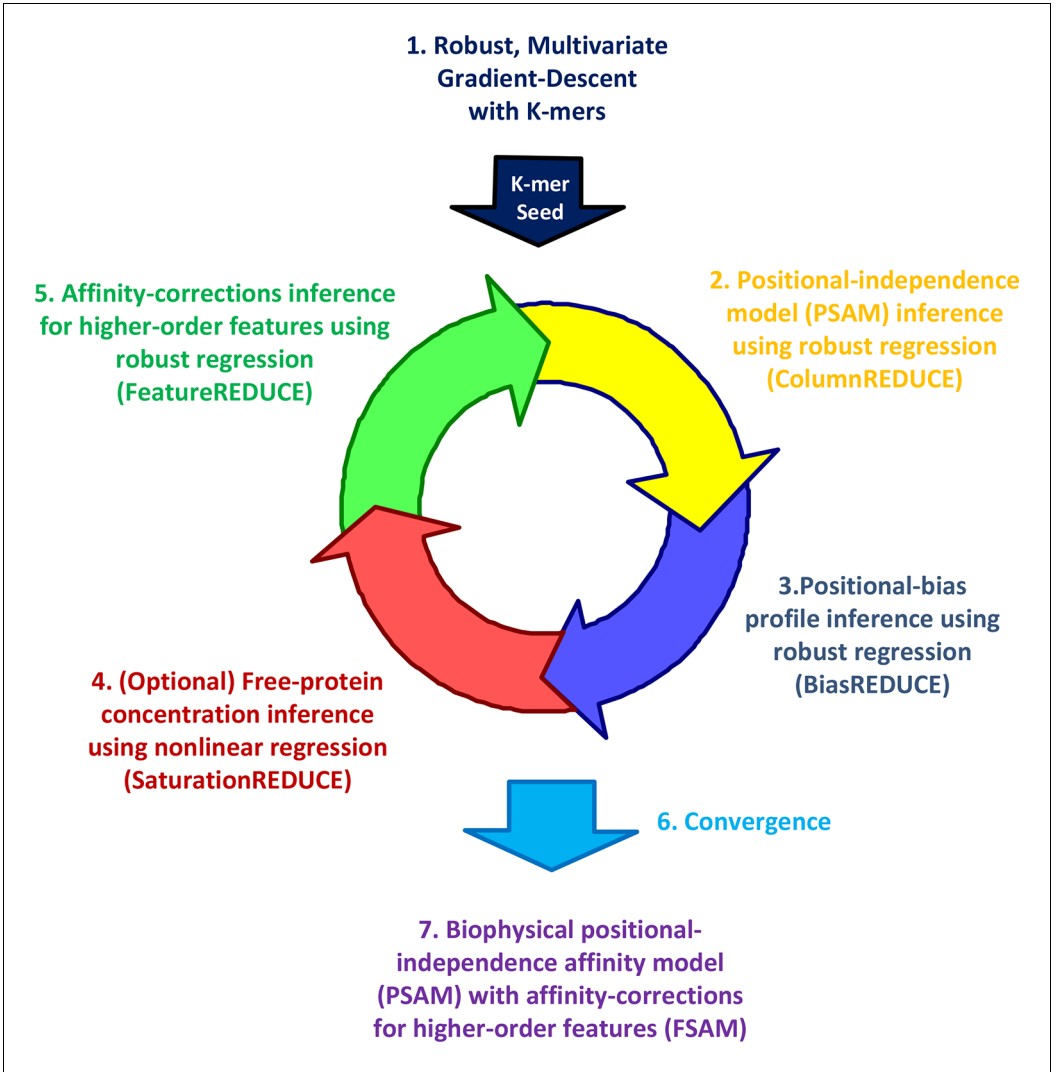

**Figure 1.** The FeatureREDUCE workflow for analyzing PBM intensities. (**1**) A robust method is used to estimate relative affinities for each K-mer of a given length. The K-mer with the highest affinity is chosen as the seed. (**2**) Using the seed as a reference, robust linear regression is used to estimate the relative affinity parameters in each column of the position-specific affinity matrix (PSAM). (**3**) With the current affinity model, linear regression is used to estimate the positional bias profile across the probe. (**4**) An optional step uses nonlinear regression to solve for the free protein concentration. (**5**) Robust regression is used to estimate free energy contributions associated with higher-order sequence features such as dinucleotides. (**6**) Steps 2 through 5 are repeated until convergence. (**7**) The procedure results in a feature-specific affinity model (FSAM) that can be used to predict the relative affinity for any DNA sequence.

---

*Zhao and Stormo, 2011*; *Benos et al., 2002*). Furthermore, while the existence of alternative binding modes is now widely recognized, accurate quantification of their relative usage has not yet been attempted. To address these needs, we developed our *FeatureREDUCE* software. It provides a flexible framework for building sequence-to-affinity models from PBM data (*Figure 1*).

## Results and discussion

*FeatureREDUCE* is based on an extensible biophysical model in which the binding free energy difference $\triangle\triangle G(S_{ref}{\rightarrow}S)$ between an arbitrary nucleotide sequence $S$ and a reference sequence $S_{ref}$ (usually taken as the highest-affinity sequence) is defined as a sum of parameters $\triangle\triangle G_{\varphi}$ over all nucleotide sequence 'features' $\varphi$ that distinguish $S$ from $S_{ref}$ (see Materials and methods). The full

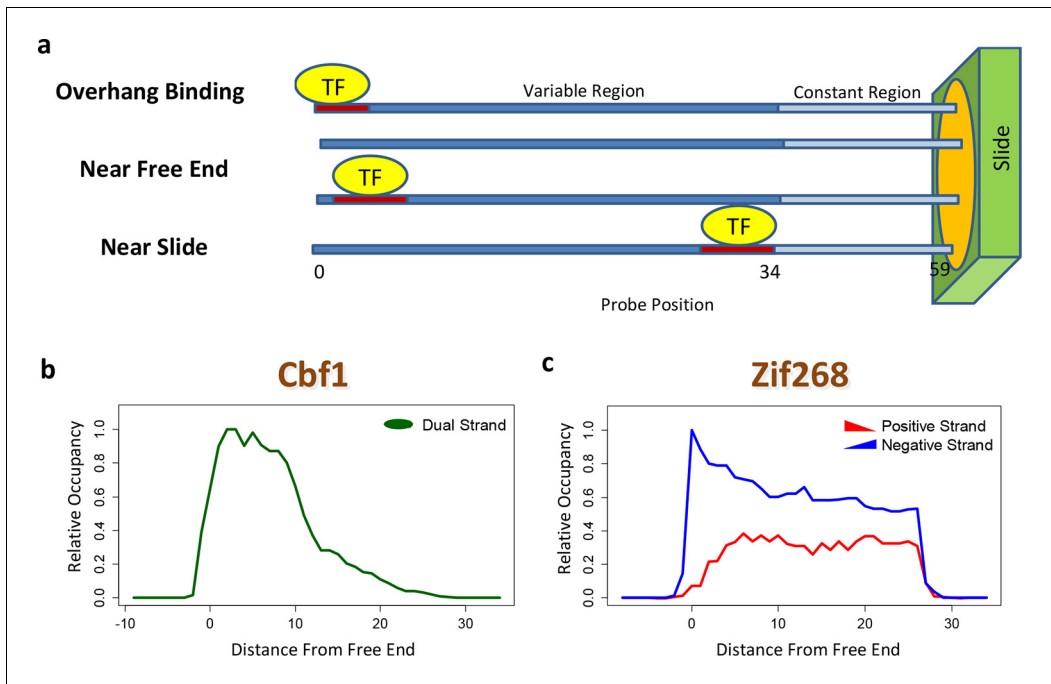

**Figure 2.** Quantifying PBM-specific positional and orientational bias. (**a**) Accounting for biases related to the position of the binding site within the probe. The effective protein concentration is lower closer to the substrate, presumably due to steric hindrance. Furthermore, binding near the free end of probe is associated with loss of contacts with the DNA backbone. (**b**) Positional bias profile for the homo-dimeric bHLH transcription factor Cbf1p, as inferred by a model fit to the PBM intensities. (**c**) Idem, for the monomeric zinc finger transcription factor Zif268. *Figure 2—figure supplement 1* shows how positional bias can be used as an indicator of data quality.

The following figure supplement is available for figure 2:

**Figure supplement 1.** Using positional bias profiles as an indicator of data quality.

set of possible features in which $S$ and $S_{ref}$ differ includes all possible single-base substitutions by default, but can be supplemented with dinucleotides that model dependencies and/or insertions at specific positions within the binding site that model variation in binding mode. Such a feature-based approach has been used previously (*Sharon et al., 2008*; *Gordân et al., 2013*; *Zhou et al., 2015*), but, as we will argue below, our approach to estimating the coefficients of the model is different and optimal.

The contribution of a binding site to the PBM intensity depends on its position within the probe, as was previously demonstrated by planting the same motif at different offsets (*Berger et al., 2006*). This may preclude accurate model estimation unless it is dealt with explicitly. Moreover, whenever the TF binds near the free end of a probe, loss of contacts with the DNA backbone can reduce binding affinity. *FeatureREDUCE* captures such spatial bias by introducing an independent multiplicative correction factor for the ratio [TF]/$K_d$ at each offset within the probe (*Figure 2a*). These coefficients are estimated from the PBM intensities in parallel with the $\triangle\triangle G$ parameters (see Materials and methods). The positional bias profile inferred by *FeatureREDUCE* for the homo-dimer Cbf1p is shown in *Figure 2b*. It indicates that the magnitude of the contribution of an individual binding site to the PBM intensity can vary by an order of magnitude depending on its offset within the probe, and that there is preference for Cbf1p binding away from the substrate. For Pho4p, binding near the free end of the probe shows an opposite trend (*Figure 2—figure supplement 1*). The fraction of the variance ($R^2$) in PBM intensity explained by a 10-bp independent-nucleotide model increases dramatically, from 48% to 71%, after accounting for the positional bias. *FeatureREDUCE* can also detect any preference for monomeric TFs to bind in one of the two possible orientations on the dsDNA probes. For example, Zif268 exhibits a strong bias for binding to the negative strand

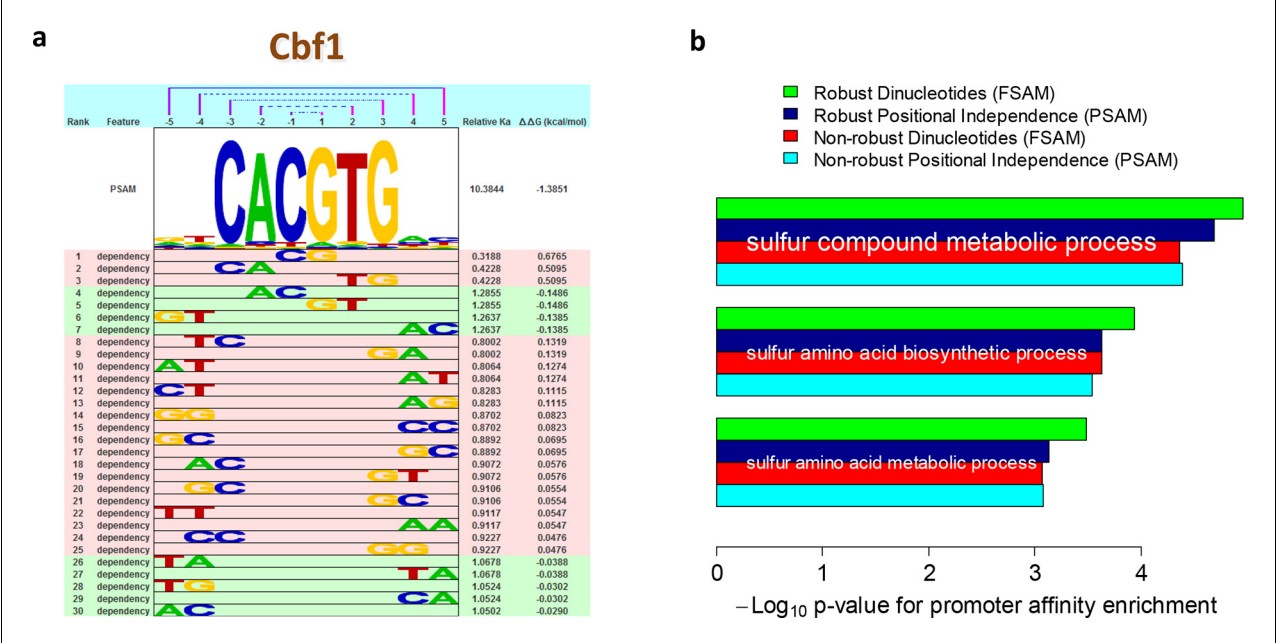

**Figure 3.** Robust estimation of dependencies between nucleotide positions. (**a**) Overview of the dependencies between pairs of neighboring nucleotides positions identified by *FeatureREDUCE* for homodimers of the basic helix-loop-helix (bHLH) factor Cbf1p. (**b**) Including dinucleotide dependencies in the sequence-to-affinity model, in combination with the use of robust regression, improves the ability to delineate Gene Ontology associations with Cbf1p targets predicted from the genome sequence. *Figure 3—figure supplement 1* shows the crucial importance of using robust inference methods for estimating the binding free energy correction terms associated with dinucleotide features. *Figure 3—figure supplement 2* shows the underlying cumulative distributions of yeast promoter affinities for 'sulfur compound metabolic process', the GO category with the most statistically significant association with Cbf1p.

The following figure supplements are available for figure 3:

**Figure supplement 1.** The crucial importance of using robust inference methods for estimating the binding free energy correction terms associated with dinucleotide features.

**Figure supplement 2.** Cumulative distributions of yeast promoter affinities for Cbf1 using four different affinity models and the GO category with the highest association p-value ('sulfur compound metabolic process').

over the positive strand (*Figure 2c*). Indeed, it is known that Zif268 requires non-specific contacts with the DNA backbone on the 5'-end of the motif (*Berger et al., 2006*). BEEML-PBM (*Zhao and Stormo, 2011*) also models positional biases along the probes, but it does not account for orientation preference, or for overhang binding at the free end of the probe.

The readout of base identity at different nucleotide positions is only approximately independent. Indeed, various studies have analyzed whether representations of sequence specificity that account for nucleotide dependencies are more accurate than those that do not (*Sharon et al., 2008*; *Agius et al., 2010*; *Lee et al., 2002*; *Stormo et al., 1986*; *Zhou and Liu, 2004*). Controversy, however, remains about whether the additional parameters associated with such dependencies reflect structural mechanisms or technology-specific biases (*Weirauch, 2013*). In the biophysical model that underlies *FeatureREDUCE*, we model dependencies by simply including additional DNA sequence features $\varphi$ that define base identity at two (or more) nucleotide positions, and estimating the corresponding free energy parameters $\triangle\triangle G_{\varphi}$ along with those for single nucleotides (see Materials and methods). The nucleotide dependencies discovered by *FeatureREDUCE* for Cbf1p are shown in *Figure 3a*. As expected for a model with additional parameters, accounting for dependencies significantly increased the fraction of the variance that could be explained when training on PBM intensities ($R^2$ improved from 71% to 96%). The real question, however, is how well the inferred model parameters perform on independent validation data.

A unique aspect of *FeatureREDUCE* is the use of robust inference techniques (*Huber and Ronchetti, 2009*), which, as it turns out, is crucial for obtaining accurate estimates of the various contributions to the binding free energy. To demonstrate this, we compared our results to measurements of binding affinity for Cbf1p obtained using the orginal version of the MITOMI technology (*Maerkl and Quake, 2007*), which are in excellent ($R^2$ = 0.95) agreement with similar measurements obtained using surface plasmon resonance (*Berger et al., 2006*; *Teh et al., 2007*). We used these 'gold-standard' binding affinity measurements to assess the quality of the sequence-to-affinity models inferred from PBM data by *FeatureREDUCE*. When we fit a position-specific affinity matrix (PSAM) based model that ignores dependencies between nucleotides, the root-mean-square error (RMSE) between *FeatureREDUCE* model predictions and gold-standard MITOMI 1.0 measurements improved from 0.071 to 0.035 when standard least-squares fitting was replaced by robust iteratively re-weighted least-squares (*Figure 3—figure supplement 1*, left panels). When positional dependencies were added to the model (feature-specific affinity model; FSAM), it actually performed worse than the model assuming independence between nucleotides (position-specific affinity matrix; PSAM) when we used *standard* least-squares fitting, indicative of over-fitting to noise in the training data. When we used *robust* regression, however, the RMSE for the FSAM-based model was significantly better than for the PSAM-based model (*Figure 3—figure supplement 1*, right panels). These results indicate that dependencies within the binding interface indeed exist, and that they can be modeled incorrectly when non-robust regression techniques are used.

We also assessed the ability of our models to make predictions regarding in vivo TF function. First, we found that the inclusion of nucleotide dependencies when predicting aggregated yeast promoter affinities improves the ability to delineate Gene Ontology categories associated with regulation by Cbf1p (see Materials and methods), but again only when robust inference techniques are used (*Figure 3B*; see also *Figure 3—figure supplement 2*). Second, to assess the extent to which we can quantitatively predict in vivo binding, we considered the degree of occupancy by Cbf1p at 955 potential genomic binding sites of type NNCACGTGNN (E-box) in yeast cells growing in rich media as measured by ChIP-seq (*Zhou and O'Shea, 2011*). Using a simple thermodynamic equilibrium model with a single free protein concentration parameter to account for binding saturation in the ChIP-seq experiment, we found that by this in vivo measure, *FeatureREDUCE* performs well (RMSE = 0.075), and significantly better than *BEEML-PBM* (*Zhao and Stormo, 2011*) (RMSE = 0.156); full data are shown in *Figure 4*.

The accuracy of *FeatureREDUCE* was demonstrated more broadly in a recent benchmark comparison between 26 different PBM data analysis algorithms (*Weirauch, 2013*). This study employed two distinct performance metrics. The in vitro metric quantified the accuracy with which probe intensities were predicted in a test PBM experiment using a model trained on independent training data from a PBM experiment for the same TF but with a different probe design; this was done for 66 different TFs from various structural families. The in vivo metric used a non-parametric score to quantify accuracy in ranking sequences in terms of their local ChIP-seq enrichment, for nine different TFs. *FeatureREDUCE* emerged as the top-performing algorithm according to both criteria.

Weirauch *et al.* (*Weirauch, 2013*) first assessed algorithm performance across a large dataset by using cross-validation between two different PBM designs. However, one of their surprising findings was that some algorithms generate models that perform well across different PBM designs but poorly when predicting in vivo binding, presumably due to over-fitting to PBM-specific biases. For example, the performance of one of the algorithms went from second-best in the PBM cross-validation test to worst in the ChIP-seq prediction test (*Weirauch, 2013*). Likewise, an extension of BEEML-PBM that adds dinucleotide parameters (*Zhao et al., 2012*) did not perform well on ChIP-seq data (*Weirauch, 2013*), presumably because it does not employ robust inference techniques. We conclude that *FeatureREDUCE* is currently the *only* algorithm that succeeds in parameterizing dependencies within the binding site.

Another current debate in the field is whether or not PBM data can provide evidence of alternative DNA binding modes employed by the same TF (*Zhao and Stormo, 2011*; *Badis et al., 2009*; *Morris et al., 2011*; *Gordân et al., 2011*). Oligomeric TF complexes can often bind DNA using different relative orientations of and/or spacing between their subunits (*Gordân et al., 2011*). The modeling framework employed by *FeatureREDUCE* provides a natural opportunity to perform forward selection of multiple binding modes, represented by distinct PSAMs, which can subsequently be combined into a single predictive model and refined in an iterative manner (see

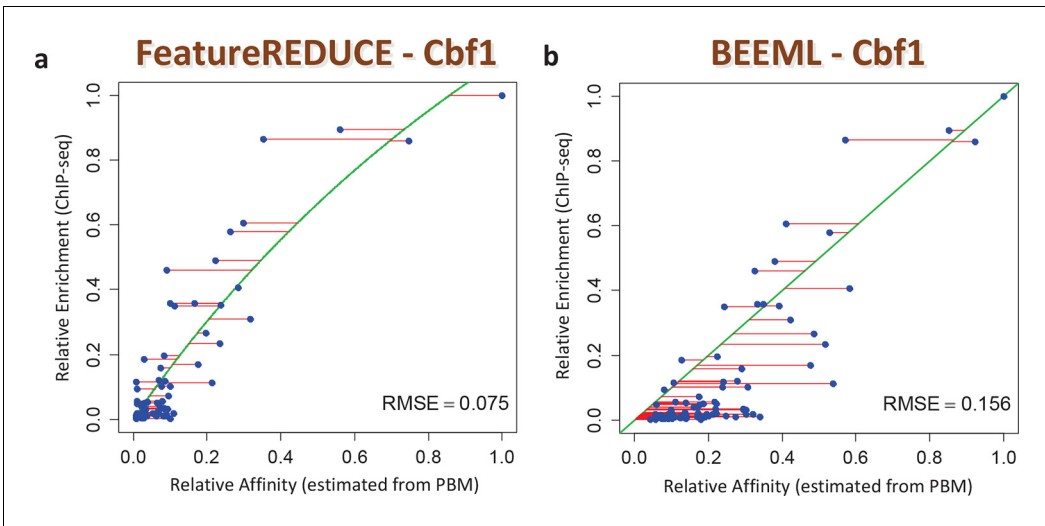

**Figure 4.** ChIP-seq based validation of position-specific affinity matrix (PSAM) inferred for Cbf1p. (**a**) Direct comparison between relative affinities for 10-mers inferred from PBM intensities by *FeatureREDUCE* and relative in vivo occupancy at 955 genomic locations of type NNCACGTGNN (E-box with flanks) as measured by ChIP-seq (*Zhou and O'Shea, 2011*). Trimmed-mean (trim = 10%) ChIP-seq fold-enrichments were computed for all unique 10-mer sequences that occur at least three times in the genome. To account for saturation of higher-affinity binding sites, a basic equilibrium model (green curve) was fit with a single free-protein parameter. Red lines indicate the error between the observed and predicted relative ChIP enrichments. (**b**) The same plot for the *BEEML-PBM* algorithm (*Zhao and Stormo, 2011*). The same equilibrium model (green curve) was fit, but the optimal free protein concentration parameter was much lower than in (**a**), so the saturation is not apparent in this case.

Materials and methods). We applied this procedure for the basic leucine zipper (bZIP) proteins Yap1p and Gcn4p. *FeatureREDUCE* accurately captures the known intrinsic differences in half-site preference of Yap1p (which prefers TTAC, typical for the C/EBP subfamily (*Warren et al., 2012*); *Figure 5a*) and Gcn4p (which prefers TGAC, typical for the CREB subfamily; *Figure 5b*). It is well known that each protein, when binding DNA as a homo-dimer, can do so with or without a 1-bp overlap between the half-sites. What is unique about our approach is that the multi-PSAM model

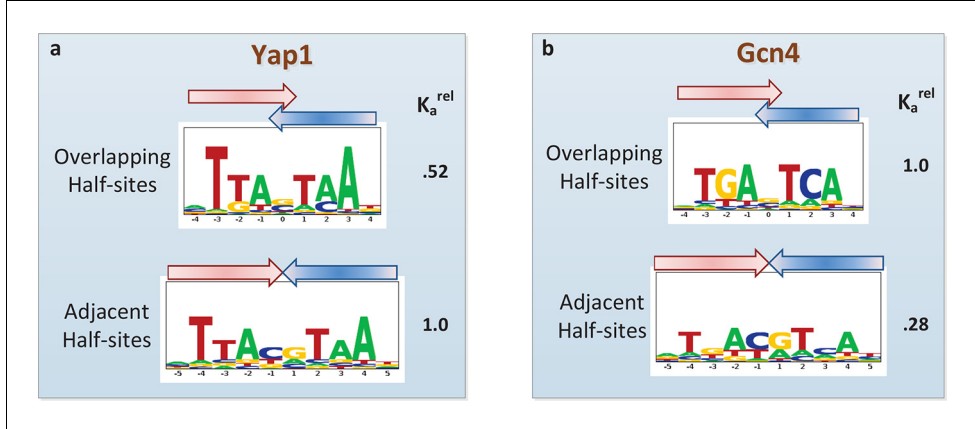

**Figure 5.** Quantifying the differential usage of alternative binding modes. The transcription factors Yap1p (**a**) and Gcn4p (**b**) can each bind in two distinct modes, in which the two half-sites respectively do (top) and do not (bottom) overlap. Not only is the sequence of preferred half-site different between the two factors, the preferred binding mode is different too, as indicated by the relative association constant ($K_a$) inferred from the PBM data by *FeatureREDUCE*.

captures fold-differences in thermodynamic stability between the binding modes by estimating an overall relative affinity coefficient for the secondary PSAM. For instance, the binding affinity of Gcn4p for TGACGTCA is predicted to be 24% of that of TGASTCA (cf. *Figure 5b*). This is in good agreement with recent high-throughput measurements of Gcn4 binding constants for all 12-mers using HiTS-FLIP (*Nutiu et al., 2011*), specifically, $K_d$ = 65 *nM* and 15 *nM* for the respective sequences. In addition, while the accuracy of the dominant TGASTCA motif essentially does not change, the accuracy of the weaker TGACGTCA motif increases significantly when using our multi-PSAM model compared to the single-fit model ($R^2$ improved from 69% to 80% for sites with relative affinity > 0.1). We note that the results of our analysis can be used in a straightforward manner to predict which binding mode is dominant for a particular DNA sequence.

In conclusion, *FeatureREDUCE* analysis yields an accurate and interpretable biophysical affinity model that can provide detailed clues about the structural mechanisms that underlie protein-DNA recognition (*Roider et al., 2007*; *Kinney et al., 2007*), such as dependencies between nucleotide positions and the modulation of binding mode by variation in the underlying DNA sequence. Our algorithm allows one to make optimal use of the large volume of data that has been generated using microarray-based protein-nucleotide interaction profiling technology.

## Materials and methods

### Equilibrium model for TF-DNA interaction

FeatureREDUCE builds upon the biophysical model for protein-DNA interaction on which Matrix-REDUCE (*Foat et al., 2006*) is based. A transcription factor *P* binds to DNA sequence *S* to form a TF-DNA complex *PS*, with forward and backward rate constants $k_{on}$ and $k_{off}$, respectively:

$$P + S \underset{k_{off}}{\overset{k_{on}}{\rightleftarrows}} PS \tag{1}$$

The affinity of P for S can be expressed in terms of an equilibrium association constant $K_a(S)$ or dissociation constant $K_d(S)$:

$$K_a(S) = \frac{1}{K_d(S)} = \frac{k_{on}}{k_{off}} = \frac{[PS]}{[P][S]} \tag{2}$$

The Gibbs free energy of binding per mole (relative to a 1M reference concentration) is given by $\Delta G$ = RT ln($K_d$ /1M) is, where R is the universal gas constant and T the absolute temperature. The fractional occupancy, *N(S)*, defined as the probability that *S* is bound by *P*, can be expressed as:

$$N(S) = \frac{[PS]}{[PS] + [S]} = \frac{[P]}{[P] + K_d(S)} = \frac{[P]K_a(S)}{[P]K_a(S) + 1} \tag{3}$$

If we assume a low-concentration regime where $[P] \ll K_d(S)$ where no saturation occurs, the expression for the occupancy simplifies to:

$$N(S) \approx \frac{[P]}{K_d(S)} = [P]K_a(S) \tag{4}$$

Relative to a reference sequence $S_{ref}$ (typically chosen to be the highest-affinity sequence), there will be multiplicative change in the affinity $K_a$, or, equivalently, an additive change $\Delta\Delta G$ in the free energy of binding, for any other sequence *S*:

$$\frac{K_a(S)}{K_a(S_{ref})} = \exp\left(-\frac{\Delta\Delta G(S)}{RT}\right) \tag{5}$$

where

$$\Delta\Delta G(S) = \Delta G(S) - \Delta G(S_{ref}) \tag{6}$$

### Feature-based model of sequence specificity

FeatureREDUCE models the relative binding free energy for sequence *S* as a sum of parameters associated with the DNA sequence features $\varphi \in \Phi(S)$ that characterize *S*:

$$\Delta\Delta G(S) = \sum_{\varphi \in \Phi(S)} \Delta\Delta G_{\varphi} \tag{7}$$

In this study, we considered both single-nucleotide features (e.g. $\varphi = A_1$, denoting the presence of an A at position 1 within the binding site window) and adjacent-dinucleotide ones (e.g., $\varphi = C_3G_4$, denoting the presence of a CpG dinucleotide starting at position 3). At a given position, exactly one of a set of 4 single-nucleotide features (A, C, G, or T) will be present in any particular sequence. We refer to such a set of mutually exclusive and jointly exhaustive features as a 'block'. Each sequence has exactly one feature from each block. A binding window of length $L$ contains $L$ mononucleotide blocks. There is a one-to-one correspondence between the (exponentiated) $\Delta\Delta G_{\varphi}/RT$ values in a mononucleotide block and a column in a position-specific affinity matrix (PSAM). Together, the dinucleotide features constitute $L-1$ dinucleotide blocks, each consisting of 16 features. Within each block, the 4 or 16 $\Delta\Delta G_{\varphi}$ values are only defined up to a common additive constant. The convention we use for mononucleotide blocks is that $\Delta\Delta G_{\varphi} = 0$ for the feature that occurs in the reference sequence. For dinucleotide blocks, however, we use a different convention intended to minimize the number of $\Delta\Delta G_{\varphi}$ values that are significantly different from zero (see below for details).

## Modeling PBM intensity

The model on which MatrixREDUCE (*Foat et al., 2006*) was based assumes that the measured fluorescence intensity $y(S)$ for probe $S$ is given by a sum over all possible ways in which the TF can bind to the probe (all possible offsets in either the forward or the reverse direction), which we will here refer to as partial 'views' $S_v$ on the full probe sequence $S$:

$$y(S) = \beta_0 + \beta_1 \sum_v e^{-\Delta\Delta G(S_v)/RT} \tag{8}$$

In FeatureREDUCE, to account for positional and directional biases in the extent to which binding affinity in a particular view contributes to the probe intensity, we introduce coefficients $\gamma_{\nu}$ that are shared across all probes:

$$y(S) = \beta_0 + \beta_1 \sum_v \gamma_v e^{-\Delta\Delta G(S_v)/RT} \tag{9}$$

We do not necessarily assume a low TF concentration (cf. *Equation 4*). Moreover, we include a term $\Delta\Delta G_{ns}$ that accounts for non-specific binding, which helps capture the DNA binding characteristics of the protein succinctly and has a positive effect on numerical convergence when estimating the model parameters. Together, this leads to the following model:

$$y(S) = \beta_0 + \beta_1 \sum_{\nu} \frac{1}{1 + (\gamma_v e^{-\Delta\Delta G(S_v)/RT} + e^{-\Delta\Delta G_{ns}/RT})^{-1}} \tag{10}$$

## Robust model parameter estimation

FeatureREDUCE estimates the parameters in *Equation 9* using iteratively reweighted least squares (IRLS). This procedure down-weights data points that have high residuals compared to the fitted model. However, the weights depend on the residuals and the residuals depend on the weights. This dependency is broken by first choosing uniform initial weights, then iteratively refitting the model and re-calculating new residuals and weights, until convergence. IRLS prevents over-fitting and thereby allows for improved parameter estimation. FeatureREDUCE uses the 'rlm' function in the MASS package for R to perform the robust regression. We set the trimmed probes hyperparameter to 20%, as we previously found this value to be optimal during cross-validation on the DREAM5 dataset (*Weirauch, 2013*). Repeated rounds of parameter re-estimation that cycle over feature 'blocks' are performed until convergence, first for mononucleotide blocks (resulting in a converged PSAM), and subsequently for dinucleotide blocks. Specifically, when estimating the free energy parameters for a given block $B$, the following model is fit:

$$y(S) = \beta_0 + \sum_{\varphi \in B} \beta_{\varphi} \sum_{v \in V_{\varphi}(S)} e^{-(\Delta\Delta G(S_v) - \Delta\Delta G_{\varphi})/RT} \tag{11}$$

Here $V_\varphi(S)$ denotes the subset of views on $S$ that contain feature $\varphi$. Note that only the $\beta_\varphi$ coefficients are treated as fit parameters here. The $\Delta\Delta G_\varphi$ values come from the previous iteration. However, they are re-estimated as:

$$\frac{\Delta\Delta G_\varphi}{RT} = -\log\left(\frac{\beta_\varphi}{\beta_{\mathrm{PSAM}}}\right) \tag{12}$$

Here $\beta_{\mathrm{PSAM}}$ stands for the value of $\beta_1$ in **Equation 12** when only mononucleotide features are fit. With this normalization, $\Delta\Delta G_\varphi$ is no longer equal to zero for dinucleotide features occurring in the reference sequence. However, most of the $\Delta\Delta G_\varphi$ values for dinucleotide features now tend to be close to zero, which is desirable. In each round, the spatial bias parameters $\gamma_v$ are also re-estimated using robust regression. After iteration and convergence over multiple such rounds, FeatureREDUCE fits the additional parameters in the non-linear model in **Equation 10** using the Levenberg-Marquardt nonlinear least-squares algorithm.

## Seed discovery

Finding a good seed for the feature-based regression procedure is a crucial first step in our algorithm. To select the seed from the set of all oligomers of length $K$, we developed a dedicated robust iterative algorithm based on trimmed means designed to deal with the sparseness of the design matrix. We fit probe intensities as a weighted sum of the number of occurrences of each of the $4^K$ oligomers:

$$y(S) = \sum_m \beta_m X_m(S) \tag{13}$$

Here $X_m(S)$ denotes the number of occurrences of oligomer $m$ in sequence $S$. Regression coefficients $\beta_m$ were initialized to a small value ($10^{-4}$) representative of non-specific binding. Next, for each individual probe $S$, we determined the coefficient value $\beta'_m(S)$ that exactly predicts the intensity:

$$\beta'_m(S) = \left(\frac{1}{X_m}\right)\left[y(S) - \sum_{m' \neq m} \beta_{m'} X_{m'}(S)\right] \tag{14}$$

We then computed the trimmed mean $\beta'_m$ (removing the top and bottom 15% values) of all $\beta'_m(S)$ values across all probes for which $X_m > 0$. Finally, we updated the coefficient according to:

$$\beta_m \longrightarrow (1-\alpha)\beta_m + \alpha\beta'_m \tag{15}$$

using a step size $\alpha = 0.1$. This was done for all oligomers in parallel. After iteration and convergence, the oligomer with the highest regression coefficient value was chosen as the seed.

## Palindromic symmetry

This step in the algorithm starts by separately fitting and then comparing positive and negative strand PSAMs. If these are similar according to the L1-norm (within a small tolerance) then the motif is flagged as symmetric. The PSAM is then rebuilt using the highest-affinity symmetric seed. The symmetric version of the binding motif tends to be more accurate while using half the number of parameters.

## Motif length

We determine the length of the binding site by adding columns to either side of the PSAM until the coefficient of determination ($R^2$) decreases. An increase indicates that there are direct or indirect protein-DNA contacts being made at the additional positions, while a decrease indicates that we have increased the length of the motif past the effective range of specificity and inadvertently excluded valid binding sites at the end of the PBM probe.

## Testing for gene ontology association

Following (*Ward and Bussemaker, 2008*), we first predict the total affinity of each gene's upstream region, as a sum over a sliding window of binding affinities computed using our model. Next, all

genes were ranked by this total promoter affinity and the Wilcoxon-Mann-Whitney rank-sum test was used to score association with each Gene Ontology category (*Gene Ontology and Consortium, 2015*). P-values were corrected for multiple testing using a Bonferroni correction based on the total number of GO categories tested in parallel.

### Alternative binding modes

To infer multiple-binding models for a single TF, the following procedure was used: (i) Fit a single-binding-model using the standard algorithm. (ii) Fit additional binding-mode model(s) to the residuals of the previous model. (iii) Iteratively update each binding mode as a weighted mean between a newly fit model and the model from the previous iteration, until convergence. (iv) Perform a final multiple regression to determine the relative preference for (and statistical significance of) each binding mode.

### Motifs containing poly-G/C stretches

Stretches of four or more guanines affect the efficiency of PBM probe synthesis and were replaced by their reverse complement in some PBM designs (*Berger et al., 2008*). Therefore, if the highest-affinity motif contains four or more consecutive cytosines, *FeatureREDUCE* uses only the positive strand to generate the biophysical model; conversely, if the highest-affinity motif contains a poly-G stretch, only the negative strand is used.

### Software availability

http://bussemakerlab.org/software/FeatureREDUCE/

## Acknowledgements

We thank the members of the Bussemaker and Mann labs for comments and feedback during the course of these studies. This work was supported by National Institute of Health grants R01HG003008 to HJB, GM058575 to RSM, F32GM087047 to MS, and F32GM099160 to NA, a John Simon Guggenheim Foundation Fellowship (HJB), and pilot funding from Columbia University's Research Initiatives in Science and Engineering (RISE) program to HJB and RSM.

## Additional information

### Funding

| Funder | Grant reference number | Author |
| --- | --- | --- |
| National Human Genome Research Institute | R01HG003008 | Harmen J Bussemaker |
| John Simon Guggenheim Memorial Foundation | | Harmen J Bussemaker |
| National Institute of General Medical Sciences | R01GM058575 | Richard S Mann |
| National Cancer Institute | U54CA121852 | Richard S Mann Harmen J Bussemaker |

The funders had no role in study design, data collection and interpretation, or the decision to submit the work for publication.

### Author contributions

TRR, HJB, Conception and design, Analysis and interpretation of data, Drafting or revising the article; AL, Conception and design, Analysis and interpretation of data; RSM, Analysis and interpretation of data, Drafting or revising the article

## Additional files

### Major datasets

The following previously published datasets were used:

| Author(s) | Year | Dataset title | Dataset URL | Database, license, and accessibility information |
|---|---|---|---|---|
| Bulyk ML, co-workers | 2006 | Transcription factor Zif268 | http://the_brain.bwh.har-vard.edu/uniprobe/de-tailsDef.php?id=400 | Publicly available at the Uni Probe Database (Accession no: UP00400). |
| Zhou X, O'Shea E | 2011 | Integrated approaches reveal determinants of genome-wide binding and function of the transcription factor Pho4 | http://www.ncbi.nlm.nih.gov/geo/query/acc.cgi?acc=GSE29506 | Publicly available at the Gene Expression Omnibus (Accession no: GSE29506). |
| Bulyk ML, co-workers | 2006 | Transcription factor Cbf1 | http://the_brain.bwh.har-vard.edu/uniprobe/de-tailsDef.php?id=397 | Publicly available at the Uni Probe Database (Accession no: UP00397). |
| Zhou X, O'Shea E | 2011 | Integrated approaches reveal determinants of genome-wide binding and function of the transcription factor Pho4 | http://www.ncbi.nlm.nih.gov/geo/query/acc.cgi?acc=GSE29506 | Publicly available at the Gene Expression Omnibus (Accession no: GSE29506). |
| Zhou X, O'Shea E | 2011 | Pho2_ChIP_HighPi | http://www.ncbi.nlm.nih.gov/geo/query/acc.cgi?acc=GSM730520 | Publicly available at the Gene Expression Omnibus (Accession no: GSM730520). |
| Maerkl, Quake | 2007 | Cbf1 MITOMI data | http://www.sciencemag.org/content/suppl/2007/01/09/315.5809.233.DC1/1131007s_database.zip | Publicly available at sciencemag.org |
| Bulyk ML, co-workers | 2009 | Transcription factor Pho4 | http://the_brain.bwh.har-vard.edu/uniprobe/de-tailsDef.php?id=332 | Publicly available at the Uni Probe Database (Accession no: UP00332). |
| Bulyk ML, co-workers | 2009 | Transcription factor Yap1 | http://the_brain.bwh.har-vard.edu/uniprobe/de-tailsDef.php?id=327 | Publicly available at the Uni Probe Database (Accession no: UP00327). |
| Bulyk ML, co-workers | 2009 | Transcription factor Gcn4 | http://the_brain.bwh.har-vard.edu/uniprobe/de-tailsDef.php?id=285 | Publicly available at the Uni Probe Database (Accession no: UP00285). |

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
