## [Decision Letter]

Thank you for sending your work entitled "Building accurate sequence-to-affinity models from high-throughput in vitro protein-DNA binding data using FeatureREDUCE" for consideration at *eLife*. Your article has been favorably evaluated by Aviv Regev (Senior editor) and three reviewers, one of whom is a member of our Board of Reviewing Editors.

The Reviewing editor and the other reviewers discussed their comments before we reached this decision, and the Reviewing editor has assembled the following comments to help you prepare a revised submission.

The main concerns raised by the reviewers were:

1) Lack of details about the methods, and in particular how dinucleotide interactions are incorporated and their impact on the quality of predictions, and the use of robust regression.

2) More thorough and detailed evaluation of the performance of the method compared to others (the reviews provide specific suggestions).

*Reviewer #1:*

During recent years there has been a wealth of information gathered about the DNA-binding specificity of proteins using microarrays and sequencing methods. A challenge posed by this data is how to generalize from experimental results to a model of the DNA-binding preferences (or affinities) of the protein. The authors describe a computational tool (FeatureREDUCE) for dealing with these challenges. They evaluate their tool on a few in vitro and in vivo datasets and compare it to recent state of the art tools.

I agree with the authors’ description of a shift in the field from data-poor regime (where only a few binding sets per protein were known) to a data-rich regime where high-throughput methods assigning affinity/preference of the protein to thousands sequences. Yet, the methodology for representing these affinities has been lagging behind the experimental advances. I also agree with their goal of finding such a representation to generalize (i.e., extrapolate) from the results of PBMs (which list affinity value for all possible 8-mers).

To the best of my understanding the new contributions here involve the following:

1) The use of feature-based "free-energy" based method combined with specific robust estimation methods. These extend previous methods of the authors by using published methods.

2) Estimating position-bias in PBM data in parallel to estimating sequence preferences.

3) The use of Poisson regression for SELEX-seq data.

The authors evaluate their claims by using existing datasets to evaluate the results of their method compared to few recent ones in the literature.

I list comments on these aspects of the paper, especially the first, below. These describe what I view as major problems in the current manuscript, but I believe they are all addressable.

The manuscript is, above all, a methods paper. It seems to be carefully designed method to solve a well-established and important problem and it definitely would be suitable for a specialized journal. For a wider audience, I would have expected a much more significant innovation that provided new and useful insights about the general computational problem and the underlying biological reality.

Major concerns:

The authors discuss at length the importance of their free-energy view of the problem. Frankly, the discussion relating free energy representation that defines a Gibbs distribution and probabilistic models (especially exponential models) has been hashed thoroughly in multiple venues and communities. In terms of representation, there is no difference between the representational powers of position-specific Affinity and position specific Probability matrix. I agree that affinities allow an elegant way to adjust the probability due to changes in protein concentration (if you believe the system is in equilibrium). However, this is a much more minute difference than the authors' presentation makes it seem. I would tone down the "sale pitch" on this view. It is a useful one to have (and has been discussed since early days, e.g., Berg and von Hippel, 1987), but it is not inherently superior.

Similarly, the introduction of feature-based representations of free-energy differences (ΔΔG) is mathematically similar or identical to previous methods (e.g., Sharon, Lubliner and Segal, 2008).

There is a great discrepancy between the main text and figures and the Materials and methods regarding the use of features (one of the novelties of the current manuscript). The discussion in the Materials and methods mentions only position-specific mutation and do not mention any higher-order feature, which ones are searched, and how. From Figure 3 assume these are strings of mutations. However, the way the method generates candidate features (all the ones I see in the figure involve pairs of adjacent positions) or search/decide among them is not specified.

Another discrepancy between the main text and the Materials and methods is the issue of robust estimation. The authors view the use of robust estimation as one of their main contributions (reference in the Abstract, dedicated supplemental figures, etc.). Yet, the Methods are vague about the details. The main text provides a reference for a book about robust statistics, which while surely a useful text, is not helpful for the reader who attempts to understand what actually happens in the estimation procedure.

Much of the evaluation of the method is based on its performance in a published comparison of many computational methods. It is surprising that one of the best performing methods in that comparison has been unpublished in the two years since the comparison was performed (the program has been available for download apparently).

The evaluation carried out in this manuscript is limited for cases where there is external information to compare against. I found the evaluation of GO enrichment p-value of Figure 3 as a very indirect (and inaccurate) indicator of quality (the difference between p-value of 10^-8^ to 10^-7^ are not that dramatic and can be driven by changes in the status of few genes). Similarly, the differences in RMSE in Figure 4 are not convincing me that the model is substantially different or better. First of all, ChIP signal (enrichment over background) is often not a reliable quantitative signal. Second, it is unclear how differences in low affinity sequences change actual predictions made with these models. I would expect an evaluation to provide the reader with a sense as to when the precise details do matter. I suspect they do, but the current presentation is focused on various summaries that are not transparent nor reflect actual differences in predictions.

*Reviewer #2:*

The manuscript describes FeatureREDUCE, a successor to the very successful MatrixREDUCE method for inferring TF binding preferences/(relative affinity) from in vitro binding data. FeatureREDUCE adds robust regression, dinucleotide affinity models, and various features specific for recent large-scale in vitro assays of TF binding (location models for PBMs, Poisson regression for sequencing-based approaches, etc.). Results are presented on a handful of TFs, so these examples simply illustrate the advantages of the new features rather than to extensively test them.

Although it seems iterative, and the validation is limited, this is very important work. The software package would be immediately useful to a large group of researchers. Despite the abundance of work in this area, there's really nothing as comprehensive (and correct) as the described software available. BEEML and BEEML-PBM come close but the manuscript nicely illustrates the advantages of FeatureREDUCE. The fact that this software works well was already illustrated in a previous study (Weirauch et al.) and all the important conceptual and engineering advances over the past few years are implemented in this package. I think *eLife* would be a great spot for this manuscript to end up.

A few things need to be fixed though. First, there are a number of missing details about the methodology that need to be filled in. Also, the manuscript is a little grandiose and needs to be toned down. Finally, and very importantly, the software needs to be open source and freely available. The value of this manuscript is the associated software; if that software isn't freely available, then there's no point in publishing this manuscript.

Major concerns:

1) It looks like in Figure 4 that you didn't permit BEEML to fit a protein concentration parameter as well, so this isn't a fair comparison. To make it fair, you could i) not fit a protein concentration parameter for FeatureREDUCE or ii) fit one for BEEML. Actually, you should do both, just to be sure.

2) Please provide more details in the "Robust Regression" about the MM-estimator. This technique will be new to most readers. How was it implemented? Are there any other free parameters besides the number of trimmed probes?

3) From Equation 11, it looks like BiasREDUCE requires non-linear regression. If so, then Figure 1 caption #3 is wrong.

4) How do you go from a seed k-mer to an initial PSAM? Presumably, you can't start with any zeros in the PSAM or you will get zero partial derivatives.

5) What is your tolerance for the L1-norm for the palindromes?

6) You need to provide more details about the multi-PSAM mode and experiments. Do you fit an initial PSAM and then fit a second one to the residuals using the entire pipeline (starting with #1)? Do you then re-fit the first PSAM? How do you decide whether or not a second PSAM is necessary?

*Reviewer #3:*

Riley and colleagues describe the development and application of a computational regression algorithm (FeatureREDUCE) to build more accurate affinity models from PBM or SELEX-seq datasets. This new method relies on iterative regression steps and the incorporation of dinucleotide dependencies to train improved affinity models for transcription factor binding. The utility of these affinity models is demonstrated on transcription factors with simple and complex binding modes.

Overall, this computational approach appears to display improved performance over other existing methods for building affinity models, and consequently will be of value to the broader scientific community. However, the current manuscript does not describe for the end-user the type of models that are generated for use in other applications – in particular how weights for the dinucleotide dependencies in the FSAM can be represented and readily displayed for interpretation. One subset of dinucleotide dependencies is displayed in Figure 3, where surprisingly (if this reviewer is interpreting the data correctly) the strongest dinucleotide contributions to the model are substantial penalties against the consensus sequence. To this reviewer, this result is counter-intuitive, and deserves comment, as it does not mesh with a simplistic view of dinucleotide effects in protein-DNA recognition (e.g. a single amino acid simultaneously and synergistically contacting neighboring base pairs in the preferred binding site). Moreover, since the authors are claiming the demonstration of the existence of dinucleotide dependencies based on their analysis, which has been a contentious point in the field, further explanation is warranted.

[Editors' note: further revisions were requested prior to acceptance, as described below.]

Thank you for resubmitting your work entitled "Building accurate sequence-to-affinity models from high-throughput in vitro protein-DNA binding data using FeatureREDUCE" for further consideration at *eLife*. Your revised article has been favorably evaluated by Aviv Regev (Senior editor), a Reviewing editor, and two reviewers.

The authors responded to the issues raised. However, the reviewers’ opinion was that this response is too terse/unsatisfactory in places. We strongly recommend that the authors attempt to provide a clear understandable description of the methods, to the extent that someone with reasonable knowledge in the field but not in this project can understand what is being done in each step.

Essential revisions:

1) The software must be deposited to a public repository (GitHub or similar). Or at the very least, the open source version of FeatureREDUCE used in this manuscript should be made available through the supplement for this journal article. Published software should have an additional guarantee of availability though the journal and/or an outside authority like GitHub (or BitBucket, or similar; we do not wish to proscribe the channel for release).

2) Clarify the methods description of discrimination from reference sequence. Reviewer comment (following the authors' response):

Again, if this reviewer is interpreting the data correctly from Figure 3, there is a penalty against CG dinucleotides at positions -1 and 1. (Likewise CA at -3,-2 and TG at 2,3) Based on the description in the Methods:

"FeatureREDUCE models the binding free energy for sequence S as a sum of coefficients associated with all the "features" that discriminate S from the reference sequence Sref (usually the DNA sequence with the highest affinity). In this study, we considered single-nucleotide features ("A at position 1") and adjacent dinucleotide features ("CG at positions 3 and 4")."

The authors describe the utilization of these features to discriminate S from Sref. The strongest (negative) dinucleotide features in Figure 3 are for dinucleotides that are in Sref, as presumably the highest affinity sequence is the reference sequence. Again, this reviewer may be missing some critical understanding of the algorithm, but based on the methods description the result does not fit with expectation. (Why would dinucleotide dependencies be discovered that apply to Sref?) Consequently, a more adequate description is required in the Materials and methods.

3) “Feature-based modeling of intensities”. This paragraph is hard to make sense of, and uses too many ill-defined terms. Please include what the features are, the coefficient (what are "columns of features"?), and how are defined (as you did for position-specific case).

4) The "Seed detection" paragraph is similarly obscure.

5) Probe-position effects – what is α? Are the γ coefficients shared among all probes? When/how are they estimated?

6) The authors use rlm (from MASS package) to estimate the "parameters in [Disp-formula equ11]". Up to now the authors discussed coefficients (are these parameters?). As far as I can tell, these do not appear in a linear form in [Disp-formula equ11]. How do you resolve that?

---

## [Author Response]

Reviewer #1:

*The manuscript is, above all, a methods paper. It seems to be carefully designed method to solve a well-established and important problem and it definitely would be suitable for a specialized journal. For a wider audience, I would have expected a much more significant innovation that provided new and useful insights about the general computational problem and the underlying biological reality.*

The technical innovations presented here make it possible to do two things that were not possible before for the first time: (i) capturing dinucleotide dependencies in a way that holds up in cross-platform validation, (ii) quantify the relative importance of iple binding modes for the same transcription factor. Indeed, Reviewer #2 states that our work is important for the genomics community, and that *eLife* will be an excellent home for it.

*The authors discuss at length the importance of their free-energy view of the problem. Frankly, the discussion relating free energy representation that defines a Gibbs distribution and probabilistic models (especially exponential models) has been hashed thoroughly in multiple venues and communities. In terms of representation, there is no difference between the representational powers of position-specific Affinity and position specific Probability matrix. I agree that affinities allow an elegant way to adjust the probability due to changes in protein concentration (if you believe the system is in equilibrium). However, this is a much more minute difference than the authors' presentation makes it seem. I would tone down the "sale pitch" on this view. It is a useful one to have (and has been discussed since early days, e.g., Berg and von Hippel, 1987), but it is not inherently superior.*

We thank the reviewer for their careful reading of the manuscript and the constructive feedback. We have revised the text to make clear that we do not claim a new representation of DNA binding specificity, but present an improved method for estimating the binding specificity parameters from high-throughput data.

We agree that the affinity and probability representations of relative base preference can often be used equivalently, as only ratios between bases are physically meaningful. However, this equivalence only holds for the probability that a given protein molecule is bound by a DNA molecule of a particular sequence. Even in this case, as the reviewer notes, they are not exactly the same, as the probabilities can be compressed relative to the affinities due to saturation at higher free protein concentrations.

Affinities always reflect the intrinsic binding preferences of the transcription factor. Base probabilities by contrast are always context dependent. This can lead to confusion and incorrect interpretation. For instance, when base probabilities are used to summarize aligned top-scoring sequences, the proportionality with affinity is lost. This is why we emphasize the difference. In fact, one could say that the essence of what FeatureREDUCE does is to first model DNA sequence probabilities in terms of TF-DNA affinities, and then put this model in reverse in order to infer the affinity parameters from the PBM data.

Nevertheless, we agree with this reviewer that we could have described the similarities and differences with alternative approaches more clearly. Therefore, as you will see, we have thoroughly rewritten the relevant section in our Introduction.

*Similarly, the introduction of feature-based representations of free-energy differences (*ΔΔ
*G) is mathematically similar or identical to previous methods (e.g., Sharon, Lubliner and Segal, 2008).*

We do agree that there is no essential difference between “energy” and “affinity” representations, as they are related by a simple log-transformation through the Boltzmann factor. We have clarified this in the text and added an additional reference (Gordon et al., 2013) to credit their early use of a feature-based representation of binding specificity.

*There is a great discrepancy between the main text and figures and the Materials and methods regarding the use of features (one of the novelties of the current manuscript). The discussion in the Materials and methods mentions only position-specific mutation and do not mention any higher-order feature, which ones are searched, and how. From Figure 3 assume these are strings of mutations. However, the way the method generates candidate features (all the ones I see in the figure involve pairs of adjacent positions) or search/decide among them is not specified.*

We agree, and thank the reviewer for bringing this to our attention. The Materials and methods now have a newly added section “Feature-based models of binding affinity” that describes in more detail how FeatureREDUCE defines dinucleotide features and estimates free energy coefficients for them.

*Another discrepancy between the main text and the Methods is the issue of robust estimation. The authors view the use of robust estimation as one of their main contributions (reference in the Abstract, dedicated supplemental figures, etc.). Yet, the Materials and methods are vague about the details. The main text provides a reference for a book about robust statistics, which while surely a useful text, is not helpful for the reader who attempts to understand what actually happens in the estimation procedure.*

Again, we agree that this is a good idea, and have provided more detail in the Materials and methods section about the iteratively reweighted least-squares (IRLS) method for robust regression.

*Much of the evaluation of the method is based on its performance in a published comparison of many computational methods. It is surprising that one of the best performing methods in that comparison has been unpublished in the two years since the comparison was performed (the program has been available for download apparently).*

We would like to emphasize that following up on the study by Weirauch et al. (2013) is only one aspect of our manuscript. We also demonstrate for the first time how the relative preference between multiple binding modes can be accurately modeled.

*The evaluation carried out in this manuscript is limited for cases where there is external information to compare against. I found the evaluation of GO enrichment p-value of Figure 3 as a very indirect (and inaccurate) indicator of quality (the difference between p-value of 10^-8^ to 10^-7^ are not that dramatic and can be driven by changes in the status of few genes). Similarly, the differences in RMSE in Figure 4 are not convincing me that the model is substantially different or better. First of all, ChIP signal (enrichment over background) is often not a reliable quantitative signal. Second, it is unclear how differences in low affinity sequences change actual predictions made with these models. I would expect an evaluation to provide the reader with a sense as to when the precise details do matter. I suspect they do, but the current presentation is focused on various summaries that are not transparent nor reflect actual differences in predictions.*

Both the ChIP and GO analysis serve to show that the affinity models are useful as predictors of in vivo function of the TF. We agree with the reviewer that the in vivo ChIP enrichment signal in general can be highly complex, but the fact that we only use the ChIP enrichment values for Cbf1p at a few hundred predicted in vitro binding sites in the genome makes this problem less severe. The GO analysis admittedly is indirect, but it is an attempt to use the affinity model to predict target genes solely from promoter sequence. This analysis was inspired by a similar one performed by Maerkl & Quake (2007) to validate their MITOMI 1.0 assay. Our affinity model is used to predict the affinity landscape across the entire upstream promoter region of each gene, integrating high and low affinity sites. As such, it puts strong demands on the affinity model. Nevertheless, not only do we find that genes in GO categories consistent with the known function of the TF tend to be associated with higher promoter affinities (and thus presumably more responsive to changes in the activity of the TF), but also that the trends regarding the benefits of robust regression and dinucleotides are in the same direction as when we validate more directly against in vitroMITOMI 1.0 data for the same TF (Figure 3—figure supplement 1). The GO and ChIP analyses are valuable complementary results in our opinion.

To address the reviewer’s concern that it is not obvious that the improvement in RMSE when dinucleotide term are added to the model in Figure 4 is statistically significant, we have added a permutation test that explicitly addresses this: we performed 100,000 iterations of randomly permuting the inferred dinucleotide corrections in Figure 4 to show that the decrease in the RMSE by a third when including dinucleotide dependencies is indeed significant (p-value 2.7e-5). Also, this significant improvement in the affinity model is not due to just one or two binding sites. Rather, as seen by the red residual lines in Figure 4, most of the binding sites show an improvement in their predicted affinities when including the dinucleotide dependencies.

We disagree with the assertion that our analysis is not transparent: we show all the data underlying our summary statistics, both for the ChIP (Figure 4) and the GO (Figure 3—figure supplement 2) analysis. The GO scoring is done using a non-parametric distributional test (Wilcoxon-Mann-Whitney), and we do not classify genes as “target” or “non-target”, and therefore the sensitivity to small changes in the status of genes does not apply here. We have added a new section to the Methods that provides these details about the GO scoring. We now apply a Bonferroni correction to the p-values, and have updated Figure panel 3B accordingly.

Reviewer #2:*A few things need to be fixed though. First, there are a number of missing details about the methodology that need to be filled in. Also, the manuscript is a little grandiose and needs to be toned down. Finally, and very importantly, the software needs to be open source and freely available. The value of this manuscript is the associated software; if that software isn't freely available, then there's no point in publishing this manuscript. It would just be free advertising for Columbia University, or Bussemaker lab, or whoever. If that's the goal, then I suggest arXiv. Ideally, the software would be put in GitHub or some other archiving service. 'Available by contacting the authors' is no longer acceptable in this field.*

We thank the reviewer for acknowledging the relevance of our work. The missing details about the methods have been added, as described below and in response to the other reviewers. We have also toned down our manuscript where was warranted.

Our software is already publicly available via our lab website. Anybody can register, and obtain it. Our reason for doing this, rather than posting it on a completely open repository such as GitHub, is that we would like to be able to email the users about future updates to the software. In response to the reviewer’s request that the software be open source, we now also include the Java source code along with the compiled classes. We have also added working examples (“demos”) for each type of analysis.

*1) It looks like in Figure 4 that you didn't permit BEEML to fit a protein concentration parameter as well, so this isn't a fair comparison. To make it fair, you could i) not fit a protein concentration parameter for FeatureREDUCE or ii) fit one for BEEML. Actually, you should do both, just to be sure.*

Although it may not be obvious, the same model with a free protein concentration parameter was fit for BEEML as for FeatureREDUCE. In the case of BEEML, the best fit was obtained with a much smaller value of this protein concentration parameter, which makes the green curve look linear. We have updated the caption to clarify this and dispel any doubts that the comparison was done fairly.

*2) Please provide more details in the "Robust Regression" about the MM-estimator. This technique will be new to most readers. How was it implemented? Are there any other free parameters besides the number of trimmed probes?*

We have provided more detail in the Methods section about the iteratively reweighted least-squares (IRLS) method for robust regression. FeatureREDUCE uses the “rlm” function in the MASS package for R to perform the robust regression. There are no other free parameters besides the number of trimmed probes. See also our response to Reviewer #1.

*3) From Equation 11, it looks like BiasREDUCE requires non-linear regression. If so, then Figure 1 caption #3 is wrong.*

We believe the caption is correct. FeatureREDUCE solves Equation 11 by cycling through the steps in Figure 1. BiasREDUCE (step 3) does not fit the free-protein concentration and thus does not need to use non-linear regression. Only SaturationREDUCE (step 4) fits the free-protein concentration and uses non-linear regression.

*4) How do you go from a seed k-mer to an initial PSAM? Presumably, you can't start with any zeros in the PSAM or you will get zero partial derivatives.*

This statement is not correct. The affinity for an entire binding site is a product over positions (or, more generally, features). Therefore, the partial derivative with respect to one of the model coefficients is a product over all other positions/features. Consequently, there is no reason for the derivative to be zero when the feature coefficient is zero (as would be the case for all three single-base features at any given position that are not in the seed).

*5) What is your tolerance for the L1-norm for the palindromes?*

The tolerance is configurable. However, the default is (0.15 x motif length). The maximum L1-distance between the positive- and negative-strand PSAMs is (2 x motifLength). We have clarified this in the Materials and methods.

*6) You need to provide more details about the multi-PSAM mode and experiments. Do you fit an initial PSAM and then fit a second one to the residuals using the entire pipeline (starting with #1)? Do you then re-fit the first PSAM? How do you decide whether or not a second PSAM is necessary?*

We now provide more detail in the Materials and methods section about the modeling of alternative binding modes. The FeatureREDUCE pipeline remains the same. However, at each step, each binding-mode model is fit to the residuals of the other models instead of the original binding data. The relative affinities between the different binding-mode models is used to determine the existence and magnitude of alternative binding.

Reviewer #3:

*Overall, this computational approach appears to display improved performance over other existing methods for building affinity models, and consequently will be of value to the broader scientific community. However, the current manuscript does not describe for the end-user the type of models that are generated for use in other applications – in particular how weights for the dinucleotide dependencies in the FSAM can be represented and readily displayed for interpretation. One subset of dinucleotide dependencies is displayed in Figure 3, where surprisingly (if this reviewer is interpreting the data correctly) the strongest dinucleotide contributions to the model are substantial penalties against the consensus sequence. To this reviewer, this result is counter-intuitive, and deserves comment, as it does not mesh with a simplistic view of dinucleotide effects in protein-DNA recognition (e.g. a single amino acid simultaneously and synergistically contacting neighboring base pairs in the preferred binding site). Moreover, since the authors are claiming the demonstration of the existence of dinucleotide dependencies based on their analysis, which has been a contentious point in the field, further explanation is warranted.*

We do not think there is anything counter-intuitive or unusual going on here, but it may require some explanation. We have therefore added some more explanation to the Materials and methods.

The dinucleotide features cannot be seen in isolation from the single-nucleotide features. To be concrete, at the two central positions, –1 and +1, we see a very strong preference for C and G, respectively, at the single-nucleotide level. When we add up the free energy coefficients for sequences that have CG at the center, with two highly favorable mononucleotide contributions and a moderately unfavorable dinucleotide contribution, the net outcome is still a very strong preference for CG over all 15 other dinucleotides. In the present paper, we are agnostic about the structural mechanisms that underlie the value of the dinucleotide coefficients, but our results are not inconsistent with one amino acid simultaneously recognizing two base pairs, the example mentioned by the reviewer.

[Editors' note: further revisions were requested prior to acceptance, as described below.]

Essential revisions: 1) The software must be deposited to a public repository (GitHub or similar). Or at the very least, the open source version of FeatureREDUCE used in this manuscript should be made available through the supplement for this journal article. This is not because we do not trust the authors. Rather, many things that can go wrong when authors are the only ones responsible for the availability of their published software. Published software should have an additional guarantee of availability though the journal and/or an outside authority like GitHub (or BitBucket, or similar; we do not wish to proscribe the channel for release).

We agree, and have created a public repository github.com/FeatureREDUCE/FeatureREDUCE/

*2) Clarify the methods description of discrimination from reference sequence. Reviewer comment (following the authors' response):*

*Again, if this reviewer is interpreting the data correctly from Figure 3, there is a penalty against CG dinucleotides at positions -1 and 1. (Likewise CA at -3,-2 and TG at 2,3) Based on the description in the Methods: "FeatureREDUCE models the binding free energy for sequence S as a sum of coefficients associated with all the "features" that discriminate S from the reference sequence Sref (usually the DNA sequence with the highest affinity). In this study, we considered single-nucleotide features ("A at position 1") and adjacent dinucleotide features ("CG at positions 3 and 4")." The authors describe the utilization of these features to discriminate S from Sref. The strongest (negative) dinucleotide features in Figure 3 are for dinucleotides that are in Sref, as presumably the highest affinity sequence is the reference sequence. Again, this reviewer may be missing some critical understanding of the algorithm, but based on the methods description the result does not fit with expectation. (Why would dinucleotide dependencies be discovered that apply to Sref?) Consequently, a more adequate description is required in the Methods.*

We agree that we did not describe how the ddG values for dinucleotide features shown in Figure 3 are normalized. For single-nucleotide features we use a standard normalization where ddG=0 for features in the optimal reference sequence. However, as we explain in detail in the revised Methods section, the normalization we use for dinucleotide features is quite different, hence the confusion on the part of the reviewer. We have also updated Figure 3 to make everything completely consistent with expectation: a newly added non-zero ddG value associated with the PSAM in Figure 3 now makes the sum over all features associated with the reference sequence equal to zero as expected. We are grateful to the reviewer for insisting on this point.

*3) “Feature-based modeling of intensities”. This paragraph is hard to make sense of, and uses too many ill-defined terms. Please include what the features are, the coefficient (what are "columns of features"?), and how are defined (as you did for position-specific case).*

We agree, and have completely rewritten this part of the Methods section.

*4) "Seed detection" paragraph is similarly obscure.*

Again we agree. We have expanded and completely rewritten this paragraph.

*5) Probe-position effects – what is α? Are the γ coefficients shared among all probes? When/how are they estimated?*

We have clarified the definition of α. Indeed, the γ coefficients are shared between all probes. We have updated the Methods section to make this clear, and also improved the description of these coefficients are defined and estimated.

*6) The authors use rlm (from MASS package) to estimate the "parameters in [Disp-formula equ11]". Up to now the authors discussed coefficients (are these parameters?). As far as I can tell, these do not appear in a linear form in [Disp-formula equ11]. How do you resolve that?*

Yes, a coefficient is a multiplicative parameter. In linear models, the parameters are therefore called coefficients. However, in nonlinear models the word parameter is more commonly used.

We agree that the procedure for solving the old [Disp-formula equ11] was not properly described, and have expanded the pertinent section in Materials and methods.